# Multi-Scale and Detail-Enhanced Segment Anything Model for Salient Object Detection

## ABSTRACT

Salient Object Detection (SOD) aims to identify and segment the most prominent objects in images. Existing methods on SOD utilize various Transformer-based models for feature extraction. However due to the scale of training datasets and training methods, these Transformer-based models still lack performance and generalization in segmentation. Segment Anything Model (SAM) is trained on a large-scale segmentation dataset, which gives it strong generalization and segmentation capabilities. Nonetheless, SAM requires accurate prompts of target objects, which is unavailable in SOD. Additionally, SAM lacks the utilization of multi-scale and multi-layer information, as well as the incorporation of fine-grained details. In order to apply SAM to SOD, and address its shortcomings, we propose a **M**ulti-scale and **D**etail-enhanced **SAM** (MDSAM). Specifically, we introduce a Lightweight Multi-Scale Adapter (LMSA), which allows SAM to learn multi-scale information with few trainable parameters. Moreover, we propose a Multi-Layer Fusion Block (MLFB) to comprehensively utilize the multi-layer information from the SAM's encoder. Finally, we propose a Detail Enhancement Module (DEM) to incorporate SAM with fine-grained details. Experimental results demonstrate the superior performance of our model on multiple SOD datasets and its strong generalization to other segmentation tasks. The source code will be publicly available.

## CCS CONCEPTS

• **Computing methodologies → Interest point and salient region detections**.

## KEYWORDS

Segment Anything Model, Salient Object Detection, Adapter, Multi-scale, Multi-layer, Detail Enhancement

## 1 INTRODUCTION

The goal of Salient Object Detection (SOD) is to detect and segment the most prominent regions in an image. SOD plays a crucial role in several downstream tasks, such as object tracking [61], object segmentation [45], and person re-identification [38].

In the last decade, Convolutional Neural Networks (CNNs) [6, 24, 39] have achieved outstanding results on SOD. However, SOD requires sufficient global semantic information, which is challenging for CNNs due to their limited receptive fields. With the strong

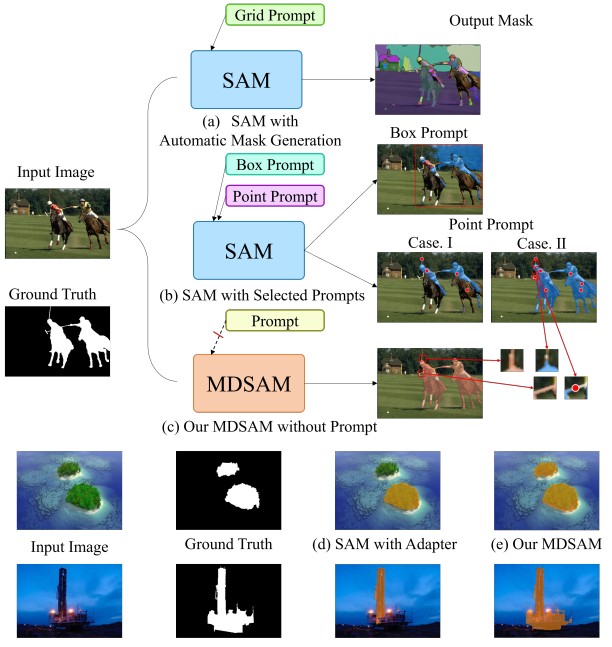

**Figure 1: (a) indicates the process of SAM automatically generating masks. (b) indicates SAM inference with carefully selected prompts. (c) shows that our MDSAM infers without prompt. (d) and (e) indicates SAM with Adapter is unable to utilize multi-scale and multi-level information. MDSAM can locate targets more accurately and segment them with fine-grained details.**

capability of global reception, Vision Transformer (ViT) [3] greatly benefits SOD that requires global object comparisons. Previous works [32, 34, 55] also use T2T-ViT [28], PVT [46], and Swin Transformer [57] for feature extraction. These Transformer-based models exhibit great performance in classification and detection, which benefit from their training on the ImageNet [35] dataset with the classification task. However, due to dataset scale and pretraining task category, the performance and generalization of the models on segmentation tasks can be further improved.

Recently, a large segmentation model called Segment Anything Model (SAM) [2] has been proposed. SAM benefits from more than 1 billion training samples, which grants it powerful generalization capability in segmentation. As a result, SAM achieves outstanding performance in various domains [21, 42, 43]. Compared to other Transformer-based foundation models, SAM is more suitable for application in segmentation tasks such as SOD. During inference, SAM requires constructive prompts such as points, boxes, or rough masks corresponding to the segmentation targets. As illustrated

by Fig. 1(a), the automatic-generated mask from SAM is hard to be used in SOD. As shown in Fig. 1(b), the point prompt requires an accurate number and placement of points. Even a slight difference can lead to incorrect results. The box prompt may be ineffective in certain scenes. Thus, applying SAM to SOD requires carefully selected prompts from targets. This is improper for SOD since the ground truth is unavailable during inference. Full fine-tuning is a method of applying SAM to SOD. However, it may lead to an excessive number of training parameters, even resulting in a decline in segmentation results. Previous works have used Adapter [33] to train large models. Nonetheless, as shown in Fig. 1(d), SAM trained with Adapter performs poorly in multi-scale scenarios. Furthermore, due to the limitation of the ViT-based encoder and the simple feature upsampling strategy in the mask decoder, SAM lacks details and causes coarse segmentation results. Additionally, in neural networks, low-level information helps the model discriminate the shape and position of the target, while high-level information aids in semantic discrimination. SAM directly utilizes the encoder's last layer, resulting in the loss of low-level information. Using the Adapter cannot solve this drawback. Fig. 1(b) illustrates that SAM encounters inaccurate segmentation and insufficient edges due to a lack of fine-grained details. As depicted in Fig. 1(d), SAM lacks the utilization of multi-layer information, leading to the model's inability to accurately discern the shape of the targets.

To address these issues, we propose a novel framework called Multi-scale and Detail-Enhanced SAM (MDSAM). MDSAM transfers SAM to the SOD task while supplying multi-scale information and fine-grained details. Specifically, we first propose a novel Lightweight Multi-Scale Adapter (LMSA). LMSA enables SAM training with fewer parameters while extracting multi-scale information. In addition, we propose a Multi-Layer Fusion Block (MLFB) to extract and fuse the outputs from different layers of the SAM's encoder. MLFB enables the decoder to fully utilize the multi-layer information. Finally, we employ a Detail Enhancement Module (DEM) to incorporate image details and edges for prediction, which helps generate precise and detailed segmentation results. MDSAM not only performs well on SOD but also exhibits superior performance on other segmentation tasks. This showcases MDSAM inherits the strong generalization capability of SAM.

Our main contributions are summarized as follows:

- We propose a novel framework for adapting SAM to SOD, named Multi-scale and Detail-enhanced SAM (MDSAM). We introduce a Lightweight Multi-Scale Adapter (LMSA) for learning task-specific information while being training-efficient and strong in acquiring multi-scale information.
- We comprehensively utilize the multi-layer information from the SAM's image encoder by using our proposed Multi-Layer Fusion Block (MLFB).
- We propose the Detail Enhancement Module (DEM) to introduce fine-grained details to segmentation results.
- We perform Extensive experiments on mainstream datasets to verify the effectiveness of our MDSAM. Further experiments are conducted to demonstrate the strong generalization of our proposed model.

## 2 RELATED WORK

### 2.1 Salient Object Detection

Currently, models for SOD are mainly divided into CNN-based and Transformer-based approaches. Due to the complexity of some scenes, which involve objects of various sizes and intricate shapes, most models consider multi-scale and multi-layer information. CNN-based models mostly use ResNet [24] as their backbone to extract multi-scale features. CPD [59] proposes a fast and accurate framework for SOD. F3Net [22] proposes a novel multi-layer fusion to solve the difference between layers. CAGNet [40] exploits the nature of multi-layer information to distinguish the salient object and suppress the non-salient regions. DFI [18] simultaneously detects salient objects, edges, and skeletons in an end-to-end unified framework. GateNet [47] addresses the lack of interference control between the encoder and the decoder as well as the disparity in contributions among different encoder blocks. MINet [51] introduces aggregate interaction modules for feature integration and self-interaction modules within each decoder unit to enhance multi-scale feature efficiency and prediction consistency. LDF [23] improves accuracy by separating the saliency map into body and detail maps and employing iterative refinement through feature interaction. MENet [49] employs a novel multi-level hybrid loss and a multi-scale feature enhancement module to improve accuracy in complex scenes. Different from the aforementioned, Tracer [30] utilizes EfficientNet [41] and enhances SOD by using attention-guided tracing modules and an adaptive pixel intensity loss function for improved performance and computational efficiency. However, CNN-based models lack the perception of long-distance information.

In contrast, Vision Transformer has a global receptive field, making it highly effective for SOD task that requires global semantic information. Nonetheless, due to the single-scale issue of ViT, Transformer-based models also adopt other models as backbones. VST [34] uses T2T-ViT [28] to capture long-range dependencies and integrates multi-level features for high-resolution results. Self-Reformer [55] employs PVT [46] as its backbone and incorporates global and local context branches to obtain both global semantic and local detail information. ICON [32] utilizes Swin Transformer [57] to extract features and enhance the integrity of detected salient regions by aggregating diverse features and improving feature channels. BBRF [31] develops a method to enhance SOD by expanding the receptive fields, which allows for more accurate detection of objects across various scales, especially those that are unusually large or small. DC-Net [19] employs a divide-and-conquer strategy with dual encoders and a novel two-level decoder to enhance SOD with high efficiency and accuracy.

Despite the impressive performance of these Transformer-based models on the SOD task, SAM exhibits superior feature extraction capabilities and robust generalization ability. Consequently, it is reasonable to transfer SAM for SOD to leverage these strengths.

### 2.2 Segment Anything Model

SAM [2] is proposed to build a foundation model for image segmentation. It performs remarkably well on many tasks [1, 42]. However, SAM requires precise prompts, such as points and boxes of the target. These prompts are difficult to obtain for SOD. Previous work [34, 55] has chosen to perform full fine-tuning on foundation

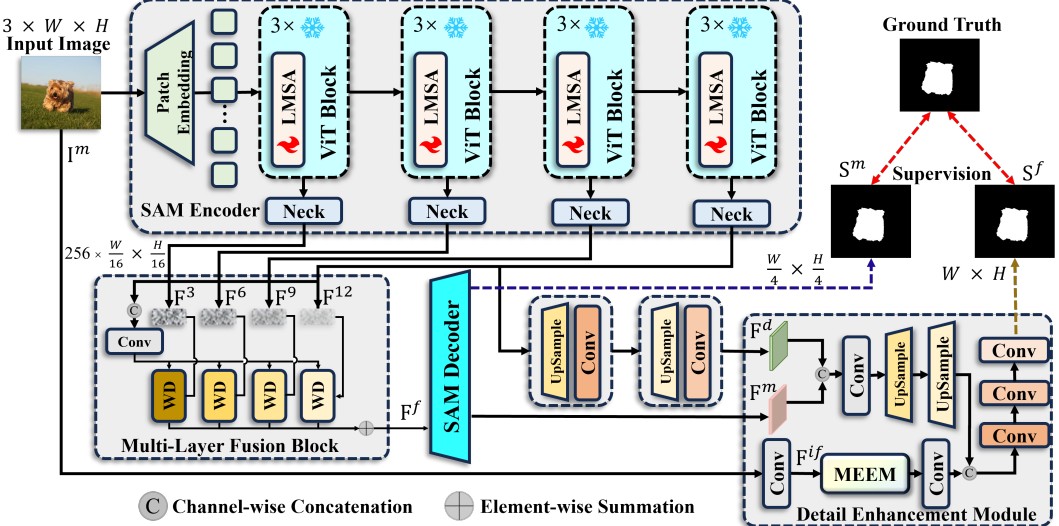

**Figure 2: Overall architecture of the proposed MDSAM. It consists of four key components: a SAM encoder with Lightweight Multi-scale Adapters (LMSA), a multi-layer Multi-Layer Fusion Block (MLFB) that includes Weight Distributors (WD), a SAM mask decoder, a Detail Enhancement Module (DEM) with Multi-scale Edge Enhancement Module (MEEM).**

models in order to transfer it to downstream tasks. Yet, direct full fine-tuning can lead to an excessive number of training parameters, even resulting in performance degradation. There are some works that attempt to transfer foundation models with a small number of trainable parameters. Adapter [33] introduces an efficient transfer learning approach for NLP by fixing the parameters of the original network and adding a few trainable parameters, achieving competitive performance with fewer parameters compared to full fine-tuning. Prefix-Tuning [27] and LoRA [13] also fix the original model parameters. Prefix-tuning prepends a sequence of continuous task-specific vectors to the input. LoRA injects trainable rank decomposition matrices into each layer of the Transformer architecture. Some works [21, 25, 43, 58] also attempt to use these methods to transfer SAM to downstream tasks.

However, these methods failed to enable SAM to learn multi-scale and multi-layer information. Furthermore, due to performance constraints, ViT-based methods require the downsampling of inputs when extracting features. This leads to a loss of detailed information. Additionally, SAM's simple decoder fails to incorporate detailed information into features, result in coarse segmentation. Our MD-SAM uses small training parameters to transfer SAM into SOD and enable SAM to acquire multi-scale information. Moreover, we introduce lightweight modules to utilize multi-layer information in the encoder and add fine-grained details to the final output.

## 3 METHOD

We propose a novel Multi-scale and Detail-Enhanced SAM (MD-SAM) for the SOD task. The overall architecture of our proposed MDSAM is shown in Fig. 2. Specifically, we describe our Lightweight Multi-Scale Adapter (LMSA) in Sec. 3.1 which reduces parameters for training and improves the multi-scale capability in semantic learning. The design of the Multi-Layer Fusion Block

(MLFB) in detail is presented in Sec. 3.2. In Sec. 3.3, we present the Detail Enhancement Module (DEM). Finally, we formulate the loss function in Sec. 3.4.

### 3.1 Lightweight Multi-Scale Adapter

Although SAM performs well on multiple segmentation tasks, the challenge of providing suitable prompts still limits its direct application in SOD, as shown in Fig. 1. One possible solution is to full fine-tuning SAM. However, the excessive number of trainable parameters from the SAM's encoder and insufficient data lead to unsatisfactory model performance. Adapter [33] and LoRA [13] are two methods for training large models with few parameters. Due to the lack of multi-scale and local information, these methods fail to enable the model to learn sufficient semantic information for SOD. To resolve these issues, we propose a Lightweight Multi-Scale Adapter (LMSA) which extracts comprehensive features by learning multi-scale information and adds depth-wise convolution at different scales to capture local information. To the best of our knowledge, we are the first to apply the Pyramid Pooling Module (PPM) [16] to acquire multi-scale information for transferring, and we make further improvements by enhancing the module's capability of extracting local information. In this way, our MDSAM incorporates multi-scale and local information while requiring few parameters for training.

As shown in Fig. 3, Each block of the SAM's encoder is composed of a Multi-Head Self-Attention (MHSA) [5], an MLP, and two normalization layers. It is expressed as follows:

$$\hat{\mathbf{X}}_i = MHSA(LN(\mathbf{X}_i)) + \mathbf{X}_i, \qquad (1)$$

$$\mathbf{X}_{i+1} = MLP(LN(\hat{\mathbf{X}}_i)) + \hat{\mathbf{X}}_i, \qquad (2)$$

where $\mathbf{X}_i \in \mathbb{R}^{N \times D}$ is the input of $i$-th Transformer block. $N$ is the number of tokens. $D$ is the Transformer's embedding dimension.

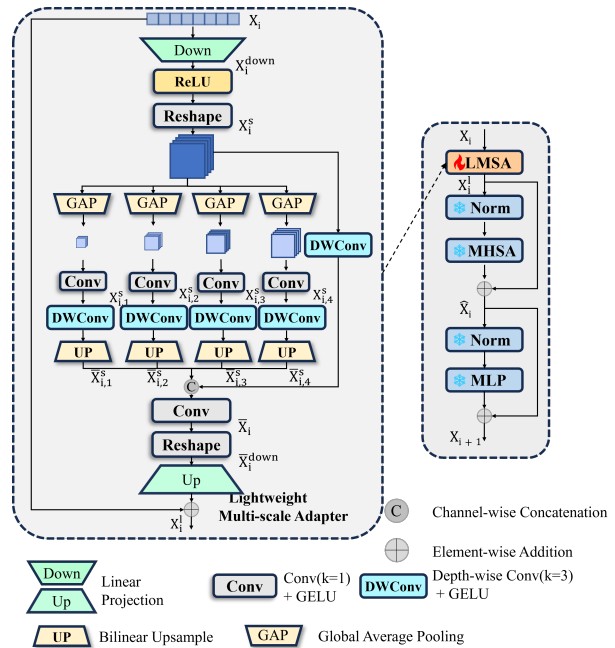

Figure 3: Details of the proposed LMSA.

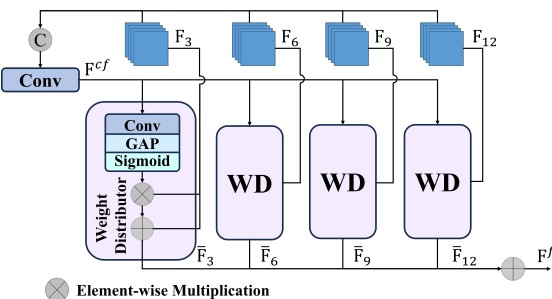

Figure 4: Architecture of the proposed MLFB.

$\hat{\mathbf{X}}_i \in \mathbb{R}^{N \times D}$ is intermediate output. *LN* denotes Layer Normalization [20]. Following the work of 3DSAM-Adapter [15], we add an LMSA before the first normalization in each Transformer block.

The detailed structure of the LMSA is shown in Fig. 3. Specifically, for a Transformer block's input $\mathbf{X}_i$, we first use a linear projection layer to reduce its dimension :

$$\mathbf{X}_i^{\text{down}} = \tau(ReLU(\mathbf{X}^{\text{down}}(\mathbf{X}_i))), \qquad (3)$$

where $\mathbf{X}^{\text{down}} \in \mathbb{R}^{D \times \frac{D}{r}}$ is a linear projection layer to reduce the number of feature channels. $\tau[\cdot]$ is the reshape operation. After a ReLU activation function, we reshape the feature into 2d feature $\mathbf{X}_i^s \in \mathbb{R}^{\frac{D}{r} \times W \times H}$ for further spatial information process.

Then, we use Global Average Pooling (GAP) layers to obtain multi-scale features $\mathbf{X}_{i,j}^s \in \mathbb{R}^{\frac{D}{4 \times r} \times W_J \times H_J}$, and utilize depth-wise convolution layers to capture local information.

$$\mathbf{X}_{i,j}^s = \phi_{1 \times 1}^l(GAP(\mathbf{X}_i^s)), 1 \le j \le 4, \qquad (4)$$

$$\bar{\mathbf{X}}_{i,j}^s = UP(DWConv(\mathbf{X}_{i,j}^s)), \qquad (5)$$

where $\phi_{1 \times 1}^l$ contains a convolution layer with $1 \times 1$ kernels and a GELU function. *DWConv* is composed of a depth-wise convolution layer with 3×3 kernels and a GELU function. *UP* denotes the bilinear interpolation for upsampling features to the specific resolution.

By upsampling features of different scales, we fuse feature $\bar{\mathbf{X}}_{i,j}^s \in \mathbb{R}^{\frac{D}{r} \times W \times H}$ with the original-scale feature $\mathbf{X}_i^s$ that contains local information after DWConv:

$$\bar{\mathbf{X}}_i = \phi_{1 \times 1}^l([\bar{\mathbf{X}}_{i,1}, \bar{\mathbf{X}}_{i,2}, \bar{\mathbf{X}}_{i,3}, \bar{\mathbf{X}}_{i,4}, DWConv(\mathbf{X}_i^s)]), \qquad (6)$$

where $\bar{\mathbf{X}}_i \in \mathbb{R}^{\frac{D}{r} \times W \times H}$ and $[\cdot]$ is channel-wise concatenation.

Finally, we reshape the feature $\bar{\mathbf{X}}_i$ to 1d feature. With a linear projection layer and residual connection [24], we get the final output $\bar{\mathbf{X}}_i \in \mathbb{R}^{N \times D}$ of LMSA:

$$\bar{\mathbf{X}}_i^{\text{down}} = \mathbf{W}^{\text{up}}(\tau(\bar{\mathbf{X}}_i)) + \mathbf{X}_i, \qquad (7)$$

where $\mathbf{W}^{\text{up}} \in \mathbb{R}^{\frac{D}{r} \times D}$ denotes a linear projection layer to restore the feature dimension.

With LMSA, SAM can be transferred to the SOD task with a small number of training parameters. Furthermore, compared to Adapter and LoRA, LMSA can better utilize multi-scale and local information with nearly the same model parameters, thereby enabling the model to learn better features.

## 3.2 Multi-Layer Fuison Block

In the SAM's encoder, each layer contains different information. Shallow layers contain more low-level information, while deep layers contain richer high-level information. High-level information is rich in semantic content, aiding the model in categorization. Low-level information includes the shape and position of objects. In the SOD task, complex scenes are often encountered. Relying solely on the high-level information from deep layers may not locate objects accurately. Therefore, leveraging multi-level information is necessary. SAM wastes multi-layer information when directly utilizing the output from the encoder's last layer as the mask decoder's input. Moreover, the simple concatenation fusion strategy cannot fully integrate the information from multiple layers. Thus, we propose a Multi-Layer Fusion Block (MLFB). As shown in Fig. 4, MLFB calculates the attention weights for different layers and obtains their proportion with Weight Distributor (WD). By using the calculated proportion, the information on each layer can be fully utilized.

We denote the features of different layers in the SAM's encoder as $\mathbf{F}_g \in \mathbb{R}^{D_1 \times H \times W}$ ($g = 3, 6, 9, 12$). $\mathbf{F}^{cf} \in \mathbb{R}^{N \times D_1}$ is generated by a simple concatenation fusion, which contains cross-layer information for weight calculation:

$$\mathbf{F}^{cf} = \phi_{1 \times 1}([\mathbf{F}_3, \mathbf{F}_6, \mathbf{F}_9, \mathbf{F}_{12}]), \qquad (8)$$

where $\phi_{1 \times 1}$ is composed of a convolution layer with $1 \times 1$ kernels, a batch normalization, and a ReLU activation function. After obtaining $\mathbf{F}^{cf}$, We send it to WD and calculate the weight of the final fusion feature for each $\mathbf{F}_g$:

$$\hat{\mathbf{P}}_g = \delta(GAP(\phi_{1 \times 1}(\mathbf{F}_g))), g \in [3, 6, 9, 12], \qquad (9)$$

$$\bar{\mathbf{F}}_g = \hat{\mathbf{P}}_g \times \mathbf{F}_g + \mathbf{F}_g, \qquad (10)$$

where $\delta$ denotes the Sigmoid function. $\hat{\mathbf{P}}_i \in \mathbb{R}^{D_1 \times 1}$ is the weight of each $\mathbf{F}_g$. Finally, we obtain the integrated feature $\mathbf{F}^f \in \mathbb{R}^{D_1 \times H \times W}$ with weighted multi-layer feature:

$$\mathbf{F}^f = \bar{\mathbf{F}}_3 + \bar{\mathbf{F}}_6 + \bar{\mathbf{F}}_9 + \bar{\mathbf{F}}_{12}. \tag{11}$$

After MLFB, $\mathbf{F}^f$ will be used as image embedding for the mask decoder. We remove the last IoU scores branch of the mask decoder and apply deep supervision for its intermediate output.

Unlike the original SAM, the features after passing through the MLFB module utilize multi-layer information. And the low-level and high-level information from encoder are fully fused. With the assistance of LMSA and MLFB, the model sufficiently leverages multi-scale and multi-level information. This greatly aids SAM's application in the SOD task.

### 3.3 Detail Enhancement Module

Despite the introduction of LMSA and MLFB aids SAM in precisely localizing targets and obtaining shapes, there are still some remaining issues. The ViT structure in SAM's encoder employs a $16 \times 16$ patch embedding strategy, which loses detailed information at high resolution. Moreover, the interpolate upsampling strategy in the SAM decoder has no detail restored, which leads to poor results quality. Thus the details and edges in SAM are not sufficiently segmented. To generate a mask with more detailed information, we propose a Detail Enhancement Module (DEM), which enhances fine-grained details at the input image resolution scale.

As shown in Fig 5, in DEM, there is a primary branch and an auxiliary branch. The primary branch upsamples features from the mask decoder output to the original input resolution. The auxiliary branch extracts fine-grained detail information from the original resolution image and adds it to the features in the primary branch. However, directly extracting details at the image input resolution would lead to excessive computation, slowing down model inference speed. Therefore, a Multi-scale Edge Enhancement Module (MEEM) is proposed. In MEEM, we use $3 \times 3$ average pooling and $1 \times 1$ convolution to replace $3 \times 3$ convolutions for the extraction of detailed information. Additionally, we utilize Edge Enhancer (EE) to strengthen edges in the feature maps.

In DEM, we first concatenate the mask decoder feature $\mathbf{F}^m$ and the last layer of encoder feature $\mathbf{F}^d$, then use a $1 \times 1$ convolution block to carry out a simple fusion. We progressively upsample the features to the input resolution by using bilinear interpolation and $3 \times 3$ convolution blocks:

$$\mathbf{F}^{si} = \phi_{1 \times 1}([\mathbf{F}^d, \mathbf{F}^m]), \tag{12}$$

$$\mathbf{F}^c = \phi_{3 \times 3}(UP_{\times 2}(\phi_{3 \times 3}(UP_{\times 2}(\mathbf{F}^{si})))) \tag{13}$$

where $UP_{\times 2}$ denotes $2\times$ bilinear interpolation upsampling. $\phi_{3 \times 3}$ contains a convolution layer with $3 \times 3$ kernels, a batch normalization, and a ReLU function. However, the coarse feature $\mathbf{F}^c$ lacks detail and edge information. Thus, we proposed a MEEM to utilize the input image to incorporate fine-grained details. Specifically, for an input image $\mathbf{F}^m$, we apply a $3 \times 3$ convolution to extract the local features of images:

$$\mathbf{F}^{if} = \phi_{3 \times 3}(\mathbf{F}^m), \tag{14}$$

where $\mathbf{F}^{if} \in \mathbb{R}^{D_2 \times H \times W}$. By the proposed MEEM, we extract edge information from the image at multiple scales obtained by different

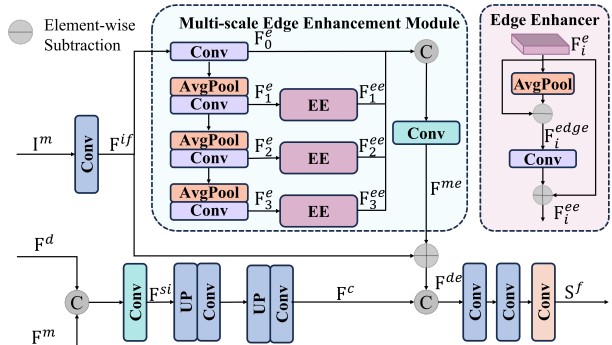

**Figure 5: Illustration of the proposed DEM.**

receptive fields, further enhancing the edge information of the features. To reduce the computational complexity, we use average pooling to expand the receptive field. The formula representation of the MEEM is as follows:

$$\mathbf{F}_0^e = \phi_{1 \times 1}(\mathbf{F}^{if}), \tag{15}$$

$$\mathbf{F}_{t+1}^e = AP(\phi_{1 \times 1}'(\mathbf{F}_t^e)), (0 \le t \le 2), \tag{16}$$

where $AP$ denotes the Average Pooling with $3 \times 3$ kernels. $\phi_{1 \times 1}'$ denotes $1 \times 1$ convolution with batch normalization and sigmoid function. $\mathbf{F}_t^e \in \mathbb{R}^{D_2 \times H \times W}$ is the features with different scale. Then, we introduce the Edge Enhancer (EE) module to strengthen the detailed information at each scale:

$$\mathbf{F}_c^{ee} = EE(\mathbf{F}_c^e), (1 \le c \le 3), \tag{17}$$

where $\mathbf{F}_c^{ee} \in \mathbb{R}^{D_2 \times H \times W}$ is the enhanced features with edge information. The operation $EE$ module is shown in the top-right part of Fig. 5, which can be represented as follows:

$$\mathbf{F}_c^{\text{edge}} = \mathbf{F}_c^e - AP(\mathbf{F}_c^e), \tag{18}$$

$$\mathbf{F}_c^{ee} = \phi_{1 \times 1}'(\mathbf{F}_c^{\text{edge}}) + \mathbf{F}_c^e, \tag{19}$$

where $\mathbf{F}_c^{\text{edge}} \in \mathbb{R}^{D_2 \times H \times W}$. Then, we fuse these features with simple concatenation and a convolution layer:

$$\mathbf{F}^{me} = \phi_{1 \times 1}([\mathbf{F}_0^e, \mathbf{F}_1^{ee}, \mathbf{F}_2^{ee}, \mathbf{F}_3^{ee}]), \tag{20}$$

where $\mathbf{F}^{me} \in \mathbb{R}^{D_2 \times H \times W}$ is the feature output of the MEEM. In this way, we obtain feature $\mathbf{F}^{me}$ that includes both the fine-grained details and the multi-scale edge information. We use these features to complement the missing information in feature $\mathbf{F}^c$. After concatenation, we apply two $3 \times 3$ convolution blocks and a $1 \times 1$ convolution block to obtain the final result $\mathbf{F}^f$:

$$\mathbf{F}^{de} = [\mathbf{F}^c, \mathbf{F}^{me} + \mathbf{F}^{if}], \tag{21}$$

$$\mathbf{F}^f = \phi_{1 \times 1}^f(\phi_{3 \times 3}(\phi_{3 \times 3}(\mathbf{F}^{de}))), \tag{22}$$

where $\phi_{1 \times 1}^f$ is a $1 \times 1$ convolution layer for output.

In DEM, the most crucial role for extracting fine-grained information for the results is the auxiliary branch. With the assistance of MEEM, MDSAM can quickly and efficiently extract rich detail information containing edges. By combining the two branches, the issue of lacking details in SAM has been resolved.

**Table 1: Quantitative comparison between our method and other SOTA methods. The best, second best, and third best results are highlighted in red, green, and blue, respectively.**

| Year | Method | Input Size | Params (M) | FLOPs (G) | FPS | DUTS-TE MAE | $F_\beta^{max}$ | $S_m$ | $E_m$ | DUT-OMRON MAE | $F_\beta^{max}$ | $S_m$ | $E_m$ | HKU-IS MAE | $F_\beta^{max}$ | $S_m$ | $E_m$ | ECSSD MAE | $F_\beta^{max}$ | $S_m$ | $E_m$ | PASCAL-S MAE | $F_\beta^{max}$ | $S_m$ | $E_m$ |
|---|---|---|---|---|---|---|---|---|---|---|---|---|---|---|---|---|---|---|---|---|---|---|---|---|---|
| | | | | | | \multicolumn{24}{c}{CNN-Based Methods} |
| 2019 | CPD [59] | 352×352 | 47.84 | 17.82 | 123 | 0.043 | 0.972 | 0.869 | 0.898 | 0.056 | 0.818 | 0.825 | 0.847 | 0.034 | 0.828 | 0.905 | 0.938 | 0.037 | 0.946 | 0.918 | 0.942 | 0.071 | 0.876 | 0.848 | 0.882 |
| 2020 | F3Net [22] | 352×352 | 25.54 | 16.48 | 167 | 0.035 | 0.905 | 0.888 | 0.920 | 0.053 | 0.841 | 0.838 | 0.864 | 0.028 | 0.943 | 0.917 | 0.952 | 0.033 | 0.957 | 0.924 | 0.948 | 0.061 | 0.892 | 0.861 | 0.898 |
| 2020 | CAGNet-L [40] | 480×480 | - | - | - | 0.029 | 0.898 | 0.897 | 0.939 | 0.047 | 0.818 | 0.845 | 0.882 | 0.024 | 0.940 | 0.923 | 0.961 | 0.026 | 0.950 | 0.930 | 0.959 | 0.063 | 0.878 | 0.870 | 0.917 |
| 2020 | DFI [18] | 224×224 | 29.61 | 11.31 | 102 | 0.039 | 0.896 | 0.887 | 0.912 | 0.055 | 0.818 | 0.839 | 0.865 | 0.031 | 0.934 | 0.920 | 0.951 | 0.035 | 0.949 | 0.927 | 0.924 | 0.065 | 0.885 | 0.857 | 0.861 |
| 2020 | GateNet-X [47] | 384×384 | 128.63 | 162.13 | 130 | 0.035 | 0.908 | 0.897 | 0.916 | 0.051 | 0.847 | 0.849 | 0.865 | 0.029 | 0.946 | 0.925 | 0.947 | 0.035 | 0.957 | 0.929 | 0.944 | 0.064 | 0.892 | 0.865 | 0.895 |
| 2020 | MINet-R [51] | 320×320 | 162.38 | 87.10 | 62 | 0.037 | 0.884 | 0.884 | 0.917 | 0.056 | 0.831 | 0.833 | 0.860 | 0.029 | 0.942 | 0.919 | 0.952 | 0.033 | 0.954 | 0.925 | 0.950 | 0.064 | 0.881 | 0.856 | 0.896 |
| 2020 | LDF [23] | 352×352 | 25.15 | 15.57 | 177 | 0.034 | 0.905 | 0.892 | 0.925 | 0.052 | 0.835 | 0.839 | 0.865 | 0.028 | 0.943 | 0.919 | 0.953 | 0.034 | 0.956 | 0.924 | 0.948 | 0.060 | 0.887 | 0.863 | 0.903 |
| 2022 | TE3 [30] | 384×384 | 14.02 | 3.23 | 24 | 0.028 | 0.909 | 0.899 | 0.943 | 0.046 | 0.840 | 0.848 | 0.881 | 0.025 | 0.944 | 0.924 | 0.961 | 0.029 | 0.954 | 0.929 | 0.958 | 0.052 | 0.896 | 0.871 | 0.916 |
| 2022 | TE5 [30] | 512×512 | 31.30 | 6.06 | 15 | 0.026 | 0.923 | 0.910 | 0.948 | 0.045 | 0.850 | 0.856 | 0.887 | 0.022 | 0.950 | 0.930 | 0.963 | 0.027 | 0.959 | 0.934 | 0.958 | 0.050 | 0.900 | 0.879 | 0.921 |
| 2022 | TE7 [30] | 640×640 | 66.27 | 10.17 | 9 | 0.023 | 0.932 | 0.920 | 0.954 | 0.045 | 0.849 | 0.856 | 0.883 | 0.021 | 0.953 | 0.934 | 0.967 | 0.026 | 0.962 | 0.936 | 0.959 | 0.047 | 0.906 | 0.883 | 0.928 |
| 2023 | MENet [49] | 354×354 | - | - | - | 0.028 | 0.918 | 0.905 | 0.938 | 0.045 | 0.845 | 0.850 | 0.871 | 0.023 | 0.951 | 0.927 | 0.960 | 0.021 | 0.957 | 0.928 | 0.951 | 0.053 | 0.897 | 0.872 | 0.910 |
| | | | | | | \multicolumn{24}{c}{Transformer-Based Methods} |
| 2021 | VST [34] | 224×224 | 44.48 | 23.18 | 70 | 0.037 | 0.895 | 0.896 | 0.919 | 0.058 | 0.836 | 0.850 | 0.871 | 0.029 | 0.946 | 0.928 | 0.952 | 0.033 | 0.954 | 0.932 | 0.951 | 0.061 | 0.882 | 0.872 | 0.902 |
| 2022 | SelfReformer [55] | 224×224 | 90.70 | 12.83 | 62 | 0.027 | 0.920 | 0.911 | 0.943 | 0.043 | 0.853 | 0.861 | 0.884 | 0.024 | 0.949 | 0.931 | 0.960 | 0.027 | 0.959 | 0.936 | 0.957 | 0.051 | 0.902 | 0.881 | 0.919 |
| 2022 | ICON-S [32] | 384×384 | 92.15 | 52.80 | 69 | 0.025 | 0.924 | 0.917 | 0.954 | 0.043 | 0.862 | 0.869 | 0.900 | 0.023 | 0.962 | 0.935 | 0.968 | 0.023 | 0.962 | 0.914 | 0.968 | 0.048 | 0.903 | 0.885 | 0.924 |
| 2023 | BBRF [31] | 352×352 | 74.00 | 67.02 | 62 | 0.025 | 0.911 | 0.909 | 0.949 | 0.044 | 0.839 | 0.861 | 0.896 | 0.020 | 0.949 | 0.932 | 0.969 | 0.022 | 0.961 | 0.939 | 0.969 | 0.049 | 0.887 | 0.878 | 0.923 |
| 2023 | DC-Net-S [19] | 384×384 | 509.61 | 211.27 | 24 | 0.023 | 0.932 | 0.925 | 0.952 | 0.039 | 0.868 | 0.875 | 0.898 | 0.021 | 0.957 | 0.941 | 0.966 | 0.023 | 0.968 | 0.947 | 0.965 | 0.049 | 0.904 | 0.887 | 0.917 |
| 2023 | SAM [2] | 512×512 | 89.94 | 103.17 | 46 | 0.030 | 0.921 | 0.909 | 0.937 | 0.044 | 0.865 | 0.869 | 0.896 | 0.022 | 0.956 | 0.935 | 0.966 | 0.025 | 0.968 | 0.944 | 0.964 | 0.061 | 0.876 | 0.866 | 0.902 |
| 2024 | **MDSAM** | 384×384 | 100.21 | 66.23 | 50 | 0.025 | 0.934 | 0.919 | 0.950 | 0.040 | 0.886 | 0.881 | 0.913 | 0.020 | 0.962 | 0.941 | 0.970 | 0.023 | 0.972 | 0.946 | 0.965 | 0.052 | 0.912 | 0.880 | 0.918 |
| 2024 | **MDSAM** | 512×512 | 100.21 | 123.44 | 35 | 0.024 | 0.937 | 0.920 | 0.949 | 0.039 | 0.887 | 0.878 | 0.910 | 0.019 | 0.963 | 0.941 | 0.969 | 0.021 | 0.974 | 0.948 | 0.967 | 0.052 | 0.907 | 0.882 | 0.917 |

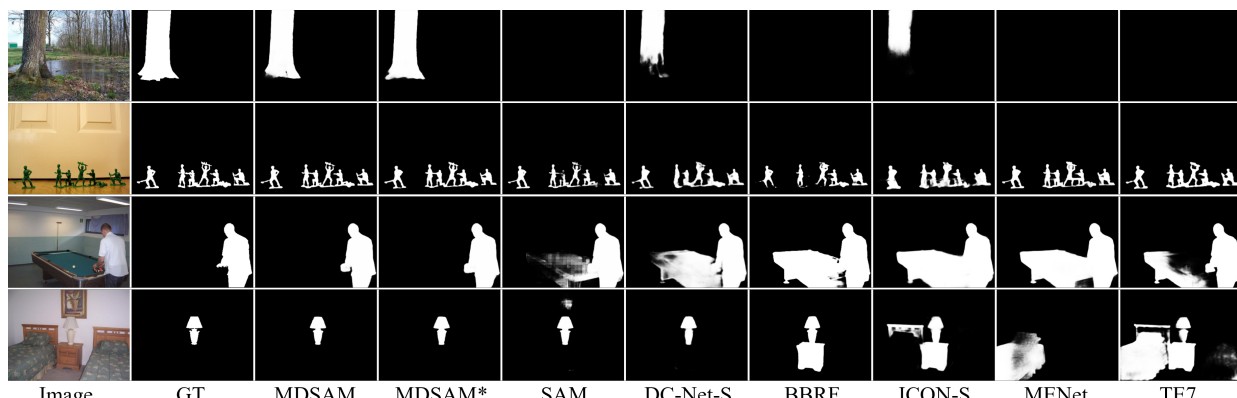

| Image | GT | MDSAM | MDSAM* | SAM | DC-Net-S | BBRF | ICON-S | MENet | TE7 |

**Figure 6: Visual comparison of saliency maps output from MDSAM and 6 other methods. MDSAM is with a $512 \times 512$ input resolution. MDSAM\* is with a $384 \times 384$ input resolution.**

## 3.4 Loss Function

Our loss functions are composed of the Binary Cross Entropy Loss, the IoU loss, and the L1 loss, which are adopted in previous methods [4, 30, 48]. However, only supervising $\mathbf{F}_f$ will cause the model excessive focus on DEM. Thus We add the same losses supervision to $\mathbf{F}_m$. The total loss of MDSAM is formulated as follows:

$$\mathcal{L}(\mathbf{S}, \mathbf{S}_{gt}) = \mathcal{L}_{BCE} + \mathcal{L}_{IoU} + \mathcal{L}_{L1}, \quad (23)$$

$$\mathcal{L}_{total} = \mathcal{L}_f(\mathbf{S}_f, \mathbf{S}_{gt}) + \mathcal{L}_m(\mathbf{S}_m, \mathbf{S}_{gt}). \quad (24)$$

## 4 EXPERIMENT

### 4.1 Experiment Setting

**Datasets.** We train our proposed MDSAM on DUTS-TR [29] (10533 images), and evaluate it on five SOD benchmark datasets, including DUTS-TE [29] (5019 images), DUTS-OMRON [7] (5168 images), HKU-IS [26] (4447 images), ECSSD [37] (1000 images) and PASCAL-S [50] (850 images).

**Metrics.** We evaluate four widely-used metrics and compare our results with state-of-the-art models. Following previous works, we calculate the Mean Absolute Error ($MAE$) [14], the max F-measure ($F_\beta^{max}$) [8], the S-measure ($S_m$) [12] and the mean Enhanced-alignment Measure ($E_m$) [11] for evaluation.

**Implementation Details.** We use an NVIDIA A100 GPU with 80 GB of memory to train our model. For initialization, we load the weights of the image encoder and mask decoder from the SAM-B model. And the rest of MDSAM is initialized randomly. We selected $512 \times 512$ and $384 \times 384$ as the model inputs and set the batch sizes to 16 and 32, respectively. We train the model using the AdamW optimizer with a weight decay of $1e^{-4}$. We freeze SAM's encoder and set the learning rate to $5e^{-5}$ for the rest of the pre-trained weights. For our proposed modules, we set the learning rate to $5e^{-4}$. We employ a warm-up period of 5 epochs and train until the maximum of 80 epochs.

### 4.2 Comparision to the State-of-the-art Methods

We compare our proposed MDSAM with 15 other models, including CPD [59], F3Net [22], CAGNet [40], DFI [18], GateNet [47], MINet [51], LDF [23], ICON [32], TE [30], MENet [49], VST [34],

**Table 2: Ablation studies of the LMSA. * denotes the number of parameters for SAM's mask decoder**

| Method | Trainable Parameters (M) | DUTS-TE | | | DUT-OMRON | | |
|--------|--------------------------|---------|---------|-------|-----------|---------|-------|
| | | $MAE$ | $F_\beta^{max}$ | $S_m$ | $MAE$ | $F_\beta^{max}$ | $S_m$ |
| Full fine-tuning | 83.43 + 3.51* | 0.030 | 0.921 | 0.909 | 0.044 | 0.865 | 0.869 |
| Adapter [33] | 7.09 + 3.51* | 0.028 | 0.923 | 0.915 | 0.045 | 0.866 | 0.871 |
| LoRA [13] | 7.09 + 3.51* | 0.028 | 0.924 | 0.914 | 0.044 | 0.864 | 0.872 |
| LMSA | 7.15 + 3.51* | **0.027** | **0.927** | **0.917** | **0.043** | **0.872** | **0.874** |

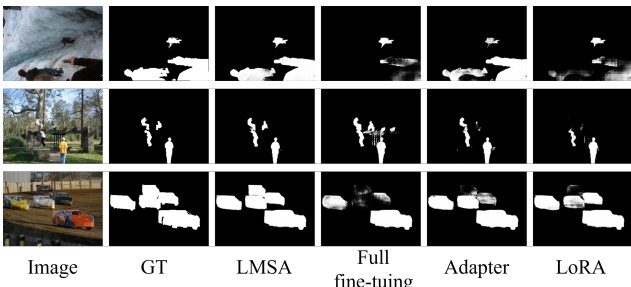

| Image | GT | LMSA | Full fine-tuing | Adapter | LoRA |

**Figure 7: Visual comparison of four SAM training strategies.**

**Table 3: Ablation studies of our proposed module. MLFB* indicates using $F^{cf}$ as the output of MLFB. DEM* indicates delete MEEM from DEM. The best scores are marked in bold.**

| Method | DUTS-TE | | | DUT-OMRON | | |
|--------|---------|---------|-------|-----------|---------|-------|
| | $MAE$ | $F_\beta^{max}$ | $S_m$ | $MAE$ | $F_\beta^{max}$ | $S_m$ |
| (a) Full fine-tuning | 0.030 | 0.921 | 0.909 | 0.044 | 0.865 | 0.869 |
| (b) SAM+LMSA | 0.027 | 0.927 | 0.917 | 0.043 | 0.872 | 0.874 |
| (c) SAM+LMSA+MLFB* | 0.027 | 0.928 | 0.918 | 0.042 | 0.871 | 0.873 |
| (d) SAM+LMSA+MLFB | 0.025 | 0.931 | 0.920 | 0.041 | 0.876 | 0.878 |
| (e) SAM+LMSA+MLFB+DEM* | 0.025 | 0.932 | **0.921** | 0.041 | 0.878 | 0.877 |
| (f) SAM+LMSA+MLFB+DEM | **0.024** | **0.937** | 0.920 | **0.039** | **0.887** | **0.878** |

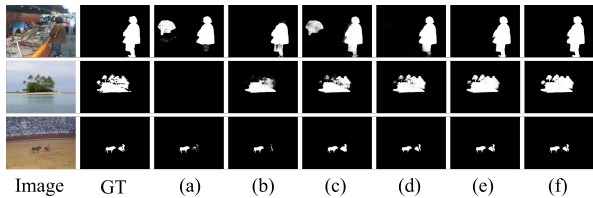

| Image | GT | (a) | (b) | (c) | (d) | (e) | (f) |

**Figure 8: Visual comparison of our proposed modules.**

SelfReformer [55], DC-Net [19], BRRF [31], SAM [2]. In our experiment, SAM uses the original design and full fine-tuning on SOD. For a fair comparison, the saliency maps are either provided by authors or generated by their released pre-trained model. And all metrics are calculated by the same tool.

**Quantitative Evaluation.** Table. 1 and Table. 2 shows the quantitative results of the compared methods. MDSAM in $512 \times 512$ input resolution achieves the best results in DUTS-OMRON, HKU-IS, and ECSSD. Furthermore, MDSAM demonstrates high competitiveness on the DUTS as well. Although MDSAM may exhibit subpar performance on the PASCAL-S dataset, it achieves the best overall results. At the $384 \times 384$ input resolution, MDSAM attains the best overall performance around similar resolutions. In Table. 1 and Table. 2, SAM is fully fine-tuned at a resolution of $512 \times 512$. Due to the lightweight designs of LMSA, MLFB, and DEM, it can be observed that compared to the original SAM, MDSAM only slightly increases the model parameters. At the same resolution, MDSAM's inference speed also only experiences a slight decrease. Moreover, when MD-SAM infers at a resolution of $384 \times 384$, it outperforms SAM at a resolution of $512 \times 512$ in both inference speed and accuracy.

**Qualitative Evaluation.** We selected four representative images for comparison. The quantitative results in Fig. 6 illustrate that in complex scenarios, MDSAM can accurately locate objects of various sizes and fully recognize the shape of objects. Furthermore, the results of our proposed MDSAM display more fine-grained details and accurate edges than other methods.

## 4.3 Ablation Study

To verify the effectiveness of our proposed modules, we conduct ablation studies with an input resolution of $512 \times 512$.

**Effectiveness of LMSA.** We conduct experiments on the original SAM structure which only deletes the prompt encoder. Following the same structures of [43, 58], which transfer SAM to downstream segmentation tasks, we introduce Adapter and LoRA to SAM for transferring to SOD. We keep the parameters of the Adapter, LoRA similar to LMSA. As shown in Table. 3, it can be observed that LMSA, with trainable parameters in SAM's encoder similar to Adapter and LoRA but significantly fewer than full fine-tuning, results in improved performance. The visual comparison in Fig. 7 further illustrates that the utilization of multi-scale information by LMSA enables the model to acquire sufficient semantic information. Consequently, it can accurately locate objects of varying sizes and quantities in complex scenarios.

**Effectiveness of MLFB.** As the results shown in Table 4's 1-3 rows when we only use $\mathbf{F}^{cf}$ (which is from a naive concatenation fusion), the performance improvement is marginal. This indicates that insufficient fusion can introduce additional noisy information to the features, thereby limiting the performance of the model. However, when we employ MLFB for fusion, there is a significant improvement compared to the absence of a fusion strategy. As shown in Fig. 8, the simple concatenation fusion strategy may confuse the model, leading to incorrect judgments. Compared to naive concatenation fusion and no fusion module, MLFB better recognizes the shape and contour of the entire object. This demonstrates that multi-layer fusion ensures the full utilization of information from each layer.

**Effectiveness of DEM.** The validation results for the DEM module and its MEEM component are shown in Table 4's 4-5 rows. The use of DEM without the MEEM module leads to slightly improved model performance, but the model still lacks sufficient edge information. However, incorporating the MEEM module results in the strongest model performance, demonstrating that the use of MEEM within the DEM enables the model to capture more detailed information. Fig. 8 illustrates that with DEM containing MEEM,

**Table 4: Performance comparison between our method and other SOD and COD methods on COD10K, CAMO, and NC4K datasets. The best, second best, and third best results are highlighted in red, green, and blue, respectively. MDSAM is with a $512 \times 512$ input. MDSAM* is with a $384 \times 384$ input.**

| Year | Method | COD10K | | | | NC4K | | | | CAMO | | | |
|------|--------|--------|---|---|---|------|---|---|---|------|---|---|---|
| | | MAE | $F_\beta^m$ | $S_m$ | $E_m$ | MAE | $F_\beta^m$ | $S_m$ | $E_m$ | MAE | $F_\beta^m$ | $S_m$ | $E_m$ |
| | | Salient Object Detection | | | | | | | | | | | |
| 2020 | F3Net [22] | 0.051 | 0.593 | 0.739 | 0.795 | 0.070 | 0.689 | 0.767 | 0.793 | 0.109 | 0.616 | 0.711 | 0.741 |
| 2020 | MINet-R [51] | 0.042 | 0.657 | 0.770 | 0.859 | 0.056 | 0.764 | 0.812 | 0.887 | 0.090 | 0.691 | 0.748 | 0.838 |
| 2021 | VST [34] | 0.042 | 0.653 | 0.781 | 0.837 | 0.050 | 0.771 | 0.831 | 0.877 | 0.076 | 0.738 | 0.787 | 0.838 |
| | | Camouflaged Object Detection | | | | | | | | | | | |
| 2021 | MGL-R [36] | 0.035 | 0.711 | 0.814 | 0.852 | 0.052 | 0.782 | 0.833 | 0.867 | 0.088 | 0.726 | 0.775 | 0.812 |
| 2021 | C2FNet [54] | 0.036 | 0.723 | 0.813 | 0.890 | 0.049 | 0.795 | 0.838 | 0.897 | 0.080 | 0.762 | 0.796 | 0.854 |
| 2021 | SINet-v2 [10] | 0.037 | 0.718 | 0.815 | 0.887 | 0.048 | 0.805 | 0.847 | 0.903 | 0.070 | 0.782 | 0.820 | 0.882 |
| 2022 | BSA-Net [17] | 0.034 | 0.738 | 0.818 | 0.891 | 0.048 | 0.808 | 0.841 | 0.897 | 0.079 | 0.763 | 0.794 | 0.851 |
| 2022 | BGNet [53] | 0.033 | 0.753 | 0.831 | 0.901 | 0.044 | 0.820 | 0.851 | 0.907 | 0.073 | 0.789 | 0.812 | 0.870 |
| 2022 | ZoomNet [52] | 0.029 | 0.766 | 0.838 | 0.888 | 0.043 | 0.818 | 0.853 | 0.896 | 0.066 | 0.794 | 0.820 | 0.878 |
| 2023 | FEDER [9] | 0.031 | 0.751 | 0.822 | 0.900 | 0.044 | 0.824 | 0.847 | 0.907 | 0.071 | 0.781 | 0.802 | 0.867 |
| 2023 | FSPNet [60] | 0.026 | 0.769 | 0.851 | 0.895 | 0.035 | 0.843 | 0.879 | 0.915 | 0.050 | 0.830 | 0.856 | 0.899 |
| 2024 | **MDSAM*** | 0.028 | 0.778 | 0.839 | 0.905 | 0.040 | 0.837 | 0.864 | 0.910 | 0.056 | 0.822 | 0.841 | 0.888 |
| 2024 | **MDSAM** | 0.025 | 0.803 | 0.862 | 0.921 | 0.037 | 0.850 | 0.875 | 0.921 | 0.053 | 0.834 | 0.852 | 0.903 |

Image  GT  MDSAM  MDSAM*  FSP  FEDER  ZoomNet  SINetv2  BGNet  BASNet

**Figure 9: Visual comparison of output from our model with 6 other methods. MDSAM is with a $512 \times 512$ input resolution. MDSAM* is with a $384 \times 384$ input resolution.**

the model obtains more fine-grained details and segments precisely.

## 4.4 Model Generalization

SAM exhibits strong generalization capabilities. Our MDSAM not only adds multi-scale, multi-layer information, and fine-grained detail to SAM but also retains the model's generalization ability. We demonstrated this in experiments on Camouflaged Object Detection (COD). Unlike SOD, which aims to find salient objects, COD focuses on detecting objects that are difficult to detect in scenes. COD10K [10] contains 5,066 camouflaged, 1,934 non-camouflaged, 3000 background images. CAMO [44] contains 1,250 camouflaged and 1,250 non-camouflaged images. NC4K [56] contains 4,121 camouflaged images. We employed a training strategy similar to [60]. We train MDSAM with all images containing camouflaged objects in the COD10K training dataset and CAMO training datasets. We test MDSAM on the test dataset. We compared our MDSAM with SOD models F3NET [22], MINet [51], VST [34], and COD models C2FNet [54], SINetv2 [10], BSA-Net [17], BGNet [53], ZoomNet [52],

and FSPNet [60]. we use four metrics for validation. Unlike SOD, we replaced the max F-measure ($F_\beta^{max}$) with the mean F-measure ($F_\beta^m$). As shown in Table. 5, our MDSAM achieves comparable performance on the COD task. In qualitative evaluation, as shown in Fig. 9, our MDSAM demonstrates more precise localization and fine-grained details. In summary, MDSAM not only performs excellently on SOD tasks but also exhibits outstanding performance on COD tasks, demonstrating our model's excellent generalization capabilities.

## 5 CONCLUSION

In this paper, we propose MDSAM for the SOD task. By introducing LMSA into SAM's encoder, we transfer SAM to SOD and enable the model to learn multi-scale information. Furthermore, we utilize the MLFB to fuse the different layers of SAM's encoder effectively. To address the issue of lacking fine-grained details in SAM, we propose DEM. Experimental results demonstrate the effectiveness and strong generalization of our approach.

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
