# OpenReview forum: "Multi-Scale and Detail-Enhanced Segment Anything Model for Salient Object Detection"
_acmmm.org/ACMMM/2024/Conference — MM2024 Poster_

### Official Review · Reviewer_pRU7 · 2024-05-23

**Rating:** 6
**Confidence:** 3

**Summary:**

This paper proposed a method called Multi-scale and Detail-enhanced SAM (MDSAM) to improve the performance and generalization in segmentation, which consists of three key components: (1) Lightweight Multi-Scale Adapter (LMSA): the component enables SAM to learn multi-scale information with minimal trainable parameters. (2) Multi-Layer Fusion Block (MLFB): the component leverages the multi-layer information of the SAM encoder. (3) Detail Enhancement Module (DEM): it integrates SAM with fine-grained detail. With these enhancements, MDSAM demonstrates good performance on multiple SOD datasets and exhibits generalization for other segmentation tasks.

**Strengths:**

1. Introduction has clear logic and problem statements, which can effectively summarize the challenges and problems in this research field, which can effectively guide the reader into the theme and content of the paper.
2. The related work is reviewed comprehensively.
3. The description of the proposed method is relatively clear, providing the detailed flowcharts to help the reader understand the implementation details of the algorithm.
4. The authors clearly explain the key information of the experiments such as datasets, evaluation metrics, training process, and hyper-parameter setting.

**Limitations:**

1. The differences and advantages between the proposed method and the existing works are needed to be further elaborated. It is suggested to strengthen the correlation analysis and contribution description, and increase the comparison of the advantages and disadvantages of different methods.
2. The experimental results are suggested to strengthen the detailed analysis of the specific data results.

**Suitability:**

3

---

### Official Review · Reviewer_VYsU · 2024-05-24

**Rating:** 4
**Confidence:** 3

**Summary:**

Salient Object Detection (SOD) aims to identify and segment the most noticeable objects in images. While Transformer-based models are commonly used for this task, their performance and generalization can be limited. The Segment Anything Model (SAM) offers strong segmentation capabilities, but it relies on accurate prompts and lacks multi-scale detail.  To leverage SAM's strengths and address its shortcomings, the authors propose MDSAM (Multi-scale and Detail-enhanced SAM), which incorporates a lightweight multi-scale adapter, a multi-layer fusion block, and a detail enhancement module. Experiments show that MDSAM excels in SOD tasks and demonstrates strong generalization to other segmentation tasks.

**Strengths:**

- Effective Adaptation and Thorough Evaluation: The paper effectively adapts the Segment Anything Model (SAM) to the Salient Object Detection (SOD) domain and presents extensive experiments demonstrating its advantages over the standard SAM model.

- Clear Motivation and Targeted Improvements: The paper clearly articulates its motivation, proposing specific enhancements to SAM to address the challenges of multi-scale information and fine-grained detail in the context of SOD.

- Well-Written and Visually Appealing: The paper is well-written, accompanied by high-quality figures, and easy to follow.

**Limitations:**

- Limited Evidence of SAM's Advantage in SOD: The paper does not convincingly demonstrate the superiority of SAM over other non-SAM Transformer methods in the SOD domain. Even with the proposed refinements to SAM, the results only achieve parity with previous state-of-the-art methods, with a potential advantage in efficiency due to pre-training. This raises questions about the necessity of adapting SAM for SOD and doesn't align with the claims made in the introduction regarding performance and generalization improvements. Additionally, the generalization experiments lack results from training SAM with the adapter.

- Scope of SAM Improvements: The modifications to SAM primarily focus on multi-scale and fine-grained details, but the zero-shot segmentation capability is not preserved. This suggests that the proposed method is more of a SAM-based SOD model rather than a fundamental improvement to SAM itself. It would be beneficial to compare the method with recent SAM-HQ work [1], which excel in fine-grained segmentation, to assess whether they can directly perform zero-shot SOD or be enhanced with the proposed adapter.

[1] Segment Anything in High Quality. NeurIPS 2024.

**Suitability:**

2

---

### Official Review · Reviewer_W5p9 · 2024-05-24

**Rating:** 5
**Confidence:** 2

**Summary:**

This article presents an improved version of the Segment Anything Model (SAM), named Multi-scale and Detail-enhanced SAM (MDSAM), for salient object detection (SOD). The MDSAM aims to address the performance and generalization issues of existing Transformer models in SOD by introducing a lightweight multi-scale adapter (LMSA), multi-layer fusion blocks (MLFB), and a detail enhancement module (DEM). Through a series of experiments, the authors have demonstrated that MDSAM exhibits superior performance on multiple SOD datasets and is capable of generalizing well to other segmentation tasks.

**Strengths:**

1. MDSAM has made effective improvements to SAM for the SOD task, with the introduction of LMSA, MLFB, and DEM significantly enhancing the model's performance.

2. The experimental results demonstrate MDSAM's advantages on multiple standard datasets, proving its effectiveness in the SOD task.

3. The article provides detailed network structure design and loss function definitions, which are helpful for readers to understand the working principle of the model.

4. The authors commit to making the source code public, which will facilitate further research and application of the model by the community.

**Limitations:**

Although the article proposes an innovative improvement plan, it lacks in-depth theoretical analysis and discussion on why specific network structures and parameters were chosen.

**Suitability:**

2

---

### Meta-Review · Area_Chair_Ptmw · 2024-07-03

**Recommendation:** Accept (Poster)
**Confidence:** 4

**Metareview:**

All reviewers suggest to accept or weekly accept.